# Hexadecenoic Fatty Acid Positional Isomers and De Novo PUFA Synthesis in Colon Cancer Cells

**DOI:** 10.3390/ijms20040832

**Published:** 2019-02-15

**Authors:** Roberta Scanferlato, Massimo Bortolotti, Anna Sansone, Chryssostomos Chatgilialoglu, Letizia Polito, Marco De Spirito, Giuseppe Maulucci, Andrea Bolognesi, Carla Ferreri

**Affiliations:** 1Consiglio Nazionale delle Ricerche, ISOF, Area della Ricerca, 40129 Bologna, Italy; r.scanferlato@hotmail.com (R.S.); anna.sansone@isof.cnr.it (A.S.); chrys@isof.cnr.it (C.C.); 2Department of Experimental, Diagnostic and Specialty Medicine-DIMES, Alma Mater Studiorum, University of Bologna, 40126 Bologna, Italy; massimo.bortolotti2@unibo.it (M.B.); letizia.polito@unibo.it (L.P.); andrea.bolognesi@unibo.it (A.B.); 3Istituto di Fisica, Fondazione Policlinico Universitario A.Gemelli IRCSS, 00168 Roma, Italy; marco.despirito@unicatt.it (M.D.S.); giuseppe.maulucci@unicatt.it (G.M.); 4Istituto di Fisica, Università Cattolica del Sacro Cuore, 00168 Roma, Italy

**Keywords:** positional isomerism, PUFA biosynthesis, membrane remodelling, membrane lipidomics, lipidomic analysis, fluidity, functional two-photon microscopy

## Abstract

Palmitic acid metabolism involves delta-9 and delta-6 desaturase enzymes forming palmitoleic acid (9*cis*-16:1; *n*-7 series) and sapienic acid (6*cis*-16:1; *n*-10 series), respectively. The corresponding biological consequences and lipidomic research on these positional monounsaturated fatty acid (MUFA) isomers are under development. Furthermore, sapienic acid can bring to the de novo synthesis of the *n*-10 polyunsaturated fatty acid (PUFA) sebaleic acid (5*cis*,8*cis*-18:2), but such transformations in cancer cells are not known. The model of Caco-2 cell line was used to monitor sapienic acid supplementation (150 and 300 μM) and provide evidence of the formation of *n*-10 fatty acids as well as their incorporation at levels of membrane phospholipids and triglycerides. Comparison with palmitoleic and palmitic acids evidenced that lipid remodelling was influenced by the type of fatty acid and positional isomer, with an increase of 8*cis*-18:1, *n*-10 PUFA and a decrease of saturated fats in case of sapienic acid. Cholesteryl esters were formed only in cases with sapienic acid. Sapienic acid was the less toxic among the tested fatty acids, showing the highest EC_50_s and inducing death only in 75% of cells at the highest concentration tested. Two-photon fluorescent microscopy with Laurdan as a fluorescent dye provided information on membrane fluidity, highlighting that sapienic acid increases the distribution of fluid regions, probably connected with the formation of 8*cis*-18:1 and the *n*-10 PUFA in cell lipidome. Our results bring evidence for MUFA positional isomers and de novo PUFA synthesis for developing lipidomic analysis and cancer research.

## 1. Introduction

The role of lipids in health is well known and lipidomics offers very powerful tools for the investigation of lipid involvement in disease onset and progress [1,2,3]. In particular, fatty acid-based membrane lipidomics and the corresponding molecular profiles provide insights of the naturally occurring and spontaneous process of lipid assembly combined with stabilized dietary intakes and biosynthesis [3]. Fatty acids are necessary building blocks for membrane phospholipid structures, regulating biophysical properties such as fluidity, dynamic features, and membrane protein functions, and have crucial importance for tumor development and disease progression [4,5]. Eukaryotic cells need polyunsaturated fatty acids (PUFA) to form their membranes; however, in cancer cells PUFAs exert various roles. In particular, phospholipase enzymes detach *n*-6 PUFA residues from membrane phospholipids, thus starting signaling cascades for proliferation, inflammation, and other crucial cellular processes [6,7]. Fatty acids also accumulate as intracellular triglyceride depots, which are necessary both for cell energy and lipotoxicity control [8,9].

Saturated fatty acid (SFA) biosynthesis starts with palmitic acid (16:0) by fatty acid synthase (FAS), which is studied in obesity [10] and cancer development [11]. The subsequent transformation of SFA into monounsaturated fatty acid (MUFA) occurs by desaturase enzymes [12,13,14]. Desaturase enzymes convert the linear structures of SFA into the bent molecular shapes of MUFA, contributing to fluidity and permeability of the membrane bilayer. Stearoyl-CoA desaturase 1 (SCD1) being a crucial enzyme for cell proliferation, strategies for its inhibition represent a novel challenge for therapeutic approaches in cancer [15,16,17]. It is worth emphasizing that desaturase enzymes insert the double bond in a regioselective manner (i.e., delta-9 desaturase inserts the double bond always in C9–C10 of the fatty acid chain), and also in a stereoselective manner, which means the double bond to be exclusively formed in the *cis* geometry that imparts the bent molecular shape. Therefore, biosynthetically produced MUFAs, such as palmitoleic and sapienic acids, can be used to monitor the endogenous formation of the unnatural *trans* geometry by a free radical mediated isomerization process of these natural *cis* isomers [18].

The biochemical pathways to the three fatty acid families from palmitic acid are depicted in Figure 1: (a) Formation of the *n*-7 MUFA series by delta-9 desaturase (SCD1) with palmitoleic acid (9*cis*-16:1), subsequently transformed by elongation into vaccenic acid (11*cis*-18:1); (b) formation of the *n*-9 MUFA series with oleic acid (9*cis*-18:1), obtained by elongation of palmitic to stearic acid (18:0) and subsequent desaturation by delta-9 desaturase; (c) formation of the *n*-10 MUFA series, more recently added to this scenario, with the palmitic acid transformation by delta-6 desaturase to sapienic acid (6*cis*-16:1) followed by the elongation step to 8*cis*-18:1 [19,20,21,22,23]. It is worth noting that the de novo synthesis of a PUFA, sebaleic acid (5*cis*,8*cis*-18:2), can only occur with the *n*-10 fatty acid series. However, this step has never been associated to cancer cells and not much is known about the biological and metabolic consequences. Lipidomic research on MUFA positional isomers of the *n*-7, *n*-9 and *n*-10 series is under development. Cao and Hotamisligil reported palmitoleic acid for its lipokine-like activities [24]. Later, mitogen activity was also reported [25]. It is a known biomarker of desaturase activity in obesity [26] and more recently was envisaged as a marker of endogenous desaturase activity and a risk of coronary heart disease in the CAREMA cohort study [27], as well as a precursor of an active metabolite in phagocytic cells [28]. The *n*-10 MUFA sapienic acid, positional isomer of palmitoleic acid, was first detected in triglycerides of human sebum [19,20], as well as in hair, nails and sebum samples [29], discussing its presence in tissues to derive from interaction with bacterial environment. We were the first to report sapienic acid as a component of human red blood cell (RBC) membrane phospholipids and of plasma cholesteryl esters [21], and to evidence its biomarker potential in morbidly obese patients [22]. Palmitic acid and *n*-3 and *n*-6 polyunsaturated fatty acids (PUFA) are competitors for the delta-6 desaturase enzyme, and the partition of palmitic acid between delta-9 and delta-6 desaturase pathways starts to be considered as a key metabolic step in health and diseases [23]. The *n*-7, *n*-9 and *n*-10 MUFA positional isomers offer a challenge for analytical characterization, adding another *n*-9 MUFA 7*cis*-16:1 to this scenario, recently identified in foamy monocytes and proposed as a biomarker for early detection of cardiovascular disease [30]. As shown in Figure 1, it is formed by beta-oxidation of oleic acid. It is worth noting that MUFAs can be present in lipid pools as triglycerides and cholesteryl esters, as well as components of membrane phospholipids, thus influencing cell biophysical and biochemical outcomes [5,31]. In our research on fatty acid supplementations in different types of tumoral cells we evidenced the crucial role of the membrane phospholipid remodelling via phospholipase A_2_ (PLA_2_) activation (i.e., the Lands cycle) [31,32,33,34,35,36]. So far the fate of sapienic acid in cancer cells is unknown.

Based on these premises, we wanted to examine the fate of sapienic acid, the positional isomer of palmitoleic acid, in particular monitoring its metabolism to the de novo synthesized PUFA sebaleic acid, a new pathway in cancer cells. We chose the model of Caco-2 cells that derives from a colorectal neoplasia and is extensively used to study the effects of diet, toxics, nutraceuticals and pharmaceuticals in human intestinal epithelium [37]. Moreover, fatty acid supplementations are well described in this cell line for palmitic acid at 50–100 µM in comparison with oleic acid and various PUFA such as arachidonic, eicosapentaenoic (EPA) and docosahexaenoic (DHA) acids [38], as well as with other *cis* and *trans* unsaturated fatty acids [39].

Here we report the fatty acid-based lipidome analysis of Caco-2 cells treated with palmitoleic and sapienic acids at two concentrations (150 and 300 µM), following up with cell morphology, viability, apoptosis markers (such as caspase 3/7 and p38 activation), cytosolic phospholipases A_2_ (cPLA_2_) activation, and the cell fluidity status. The saturated fatty acid palmitic acid was also used for comparison. Cell fluidity was determined using two-photon fluorescent microscopy with Laurdan as a fluorescent dye, as it is highly sensitive to the presence and mobility of water molecules within the membrane bilayer by a shift in its emission spectrum; its use has been described in several contexts [40,41].

These data provide the first information on how the difference in the double bond position of two carbon atoms, such as how it occurs in positional fatty acid isomers, could induce differences of biological and biophysical properties. The overall aim of this study is to contribute to the debate on lipidomics in cancer cells providing novel information on MUFA metabolism and endogenous PUFA formation.

## 2. Results

### 2.1. Effect of C16 Fatty Acid Supplementation on Cell Viability

Caco-2 cells were treated with three fatty acid supplementations (palmitic, palmitoleic and sapienic acids) and the cell viability was evaluated at concentrations ranging from 100 to 300 µM (100, 150, 200, 250 and 300 µM) at different times up to 96 h, as shown in Figure 2A, expressing the percentage of viability compared to control cultures as mean ± SD of three different experiments. At 100 µM concentration only palmitic acid was able to affect cell viability with a range of 20–40% cell viability reduction observed in the interval of 24–96 h, becoming significant after 24 h. At 150 µM concentration, palmitic acid caused a marked reduction of cell viability that decreased to almost 50% of control values after 24 h, and became almost 5% after 48–96 h. The two MUFAs showed a marked dose–effect relationship, with significant viability reduction compared to control cells at 200 µM, about 60% for sapienic acid after 72–96 h and 80% for palmitoleic acid after 24–96 h. The highest toxic effect was reached for both fatty acids at 300 µM concentration, reducing cell viability almost to 0% for palmitoleic acid, whereas viability was not absent for sapienic acid, being reduced at 25% after 24 h and later. At low concentrations (100–200 µM) palmitoleic and sapienic acids gave a similar effect on Caco-2 cells, except for the 200 µM–72 h, condition in which sapienic acid was more toxic than palmitoleic (*p* < 0.0001). At higher concentrations (250 and 300 µM) palmitoleic acid was significantly more toxic than sapienic acid (*p* < 0.0001). The concentration of each fatty acid required to reduce the Caco-2 cell viability to 50% (EC_50_) was calculated from each dose–response curve by linear regression analysis (Table 1). After 24 h incubation, the EC_50_s of the three fatty acids were in the same concentration range (see Table 1). Instead, at 48 h and later, the EC_50_ of palmitic acid was 2–2.3-fold lower (99.6–101.1 μM) than that calculated for the two unsaturated fatty acids (palmitoleic acid: 200–214.3 μM; sapienic acid 230.2–232.3 μM).

### 2.2. Effect of C16 Fatty Acid Supplementation on Cell Morphology

The morphology of Caco-2 cells exposed to 150, 250 and 300 µM fatty acid concentrations was assessed by phase contrast microscopy after 24 h (Figure 2B). All the treatments caused marked alterations of cell morphology in comparison with control cultures. Cells treated with palmitoleic and sapienic acids presented lipid droplets into the cytoplasm due to lipid accumulation. Droplets were present at 150 µM concentration and strongly augmented when cells were treated with 250 µM fatty acids. At 300 µM concentration, beside the cytoplasmic lipid accumulation, there was evident reduction of the cell population and typical apoptotic changes in the residual cell population. Differently, cells treated with palmitic acid did not show presence of cytoplasmic droplets but they appeared in a reduced number of cell viability and with a peculiar “geometric aspect”. These morphological changes in Caco-2 cells treated with high palmitic acid concentrations were previously described by van Greevenbroek [39].

### 2.3. Evaluation of Death Pathways after C16 Fatty Acid Supplementation

To evaluate the death pathways involved in Caco-2 cell death induced by the three fatty acids, a further series of experiments was carried out. With different concentrations affecting cell viability (see Figure 2A), we chose the minimum concentration of each fatty acid causing about 75–80% reduction of cell viability after 48 h (i.e., 150 µM palmitic acid, 250 µM palmitoleic acid and 300 µM sapienic acid). Double staining with AnnexinV (AnnV)/Propidium Iodide (PI) showed that, after 24 h, 42% of Caco-2 cells treated with palmitic acid were positive only to PI and about 5% were positive to AnnV. A different behaviour was observed with the two unsaturated fatty acids: The cells positive only to PI were 4.3 and 5.5%, whereas the cells positive to AnnV were 14.6% and 18.7%, for palmitoleic and sapienic acid, respectively (Figure 3A). To demonstrate the involvement of caspase-dependent apoptosis, caspase 3/7 activation was measured in cells exposed to fatty acids for different incubation times, ranging from 8 to 24 h (Figure 3B). The data showed the activation of effector caspases, which became significant after 24 h. In contrast with the data obtained by AnnV/IP staining, the higher caspase activation was reported for palmitic acid, reaching about 300% of untreated control values. To better clarify the conflicting results obtained by the AnnV/PI staining and caspase 3/7 activity, we followed an indirect approach, by evaluating the possible protective effect on cell viability of two inhibitors of specific pathways of cell death: The pan-caspase inhibitor Z-VAD and the necroptosis inhibitor Necrostatin-1. Z-VAD irreversibly binds to the active site of caspases; Necrostatin-1 inhibits the kinase activity of Receptor Interacting Protein-1 (RIP-1). Both Z-VAD and Necrostatin-1 were able to significantly protect Caco-2 cells from death induced by fatty acids, indicating the involvement of both apoptotic and necroptotic death mechanisms (Figure 3C). We also evaluated p38 and cPLA_2_ phosphorylation in Caco-2 cell lysates following supplementation with the three fatty acids by western blot analysis at 1 h and 3 h incubation times, controlling that the total cell contents of p38 and cPLA_2_ remain almost unaltered (see Appendix A: Western blot analyses of (A) p38 and(B) cPLA2 total proteins and phosphorylated forms in Caco-2 cells after supplementation with palmitic (16:0), palmitoleic and sapienic acids). In Appendix A the band intensities of p38 and cPLA_2_ phosphorylated forms are expressed as a percentage of control samples, after normalization to the intensity of the corresponding total form. The p38 mitogen-activated protein kinases are a class of enzymes, activated by several stress stimuli, i.e., cytokines, ultraviolet irradiation, heat shock, and hyperosmolality. They are involved in several important cellular pathways, as cell differentiation, apoptosis and autophagy. Their activation via phosphorylation (p-p38) increases the enzymatic activity of several substrates, such as cPLA_2_ [42]. In its turn, the activated/phosphorylated form of cPLA_2_ (p-cPLA_2_) is known to be a central regulator of stimulus-coupled cellular arachidonic acid mobilization, exhibiting a significant selectivity toward phospholipids bearing arachidonic acid. In our experiments, we showed that cells treated with palmitic, palmitoleic and sapienic acids showed quite similar behaviour in the pathway of activation of p38 and cPLA_2_. The p38 activated forms were significantly increased, reaching a value of about 150% (after 1 h). The amount of p-p38 resulted slightly, but not significantly, decreased after 3 h of incubation (Appendix A). Phosphorylation of cPLA_2_ in Caco-2 cells supplemented with the three fatty acids showed a similar trend to p-p38, but higher amounts of phospho-forms with respect to controls were measured, reaching values of about 290–340% after 1 h and 250–300% after 3 h (Appendix A).

### 2.4. Fatty Acid-Based Membrane Lipidomic Monitoring

The monitoring of membrane fatty acids was carried out in Caco-2 cells after 0.5, 1, 3 h and at 24 h of incubation with palmitoleic, sapienic and palmitic acids at 150 µM and 300 µM. Changes were statistically evaluated in comparison with cells without supplementation at the same conditions and incubation times in triplicates. We were aware of the importance of culture-related factors for the homogeneity of cell characteristics and precise evaluation between treated and untreated cells, therefore specific care was taken in the methods and sample preparation according to literature [38]. We followed published protocols for membrane isolation and preparation of fatty acid methyl esters (FAME) for gas chromatographic (GC) analysis [22,35,36], especially regarding the hexadecenoic acid isomer separation and recognition. In the present work an accurate calibration for fatty acid quantitation was performed, and for two FAME, namely 8*cis*-18:1 and 5*cis*,8*cis*-18:2, calibration parameters were established in our conditions as reported in Appendix A. In Appendix A, all the quantitative analyses of Caco-2 cell fatty acid composition with 150 and 300 µM fatty acids in comparison with controls are collected in tables, indicating their content in µg/mL as mean values ± SEM, as follows: (a) Membrane phospholipids monitoring at 0.5, 1, 3 h (Appendix A), (b) membrane phospholipids after 24 h of treatment (Appendix A); (c) triglyceride fatty acid composition after 24 h treatment with palmitic and palmitoleic acids (Appendix A) and triglyceride fatty acid composition with sapienic acid at 0.5, 1, 3, 24 h (Appendix A); (d) cholesteryl ester fatty acid composition, a lipid class found only in cells supplemented with sapienic acid (Appendix A).

As a consequence of the c-PLA_2_ activation at short incubation times (Appendix A), the membrane phospholipid remodelling can occur according to the well-known Lands cycle [31]. Therefore, we followed the membrane fatty acid content variations in the first 3 h of incubation with 150 µM supplementation of the three fatty acids. The results are summarized in Table 2, showing all statistically significant variations of membrane fatty acid levels with respect to controls, in terms of increase (↑) and decrease (↓). Immediately after 30 min of sapienic acid supplementation the incorporation in membrane phospholipids of its elongation product 8*cis*-18:1 and, after 1 h, of the subsequent desaturation product (sebaleic acid, 5*cis*,8*cis*-18:2) was found, evidently following the metabolic pathways shown in Figure 1 (see Appendix A for the values). In the same time frame 150 µM palmitoleic acid supplementation gave different results, with an immediate and significant incorporation of this fatty acid together with the other two SFAs (palmitic acid and 20:0) in membrane phospholipids. It is worth noting that palmitic acid supplementation did not produce changes at 30 min (values are reported in Appendix A). Under 150 µM supplementation of palmitoleic acid, after 1 h, the membrane PUFA content changed with the significant diminution of two *n*-6 fatty acids (20:3 and 20:4, dihomo gamma linolenic acid and arachidonic acid, respectively) and one *n*-3 fatty acid (22:5, docosapentaenoic acid). Interestingly, the supplementation of sapienic acid at identical time and concentration gave a different outcome, with increases of arachidonic acid (20:4), *n*-10 MUFA 8*cis*-18:1 and the PUFA sebaleic acid (5*cis*,8*cis*-18:2) in membrane phospholipids, whereas *n*-3 fatty acid levels were not affected.

After 3 h the PUFA scenario in cell membrane phospholipids changed for both positional isomers, producing the diminutions of omega-6 arachidonic and linoleic acids. Total omega-6 content after 3 h was reduced in cell membranes with all the three supplementations.

In 24 h incubation with 150 µM of the three fatty acids, the membrane lipidome of Caco-2 cells was profoundly changed (Appendix A) and it is very interesting to note that in the case of sapienic acid, the membrane phospholipid content of 8*cis*-18:1 (327.2 ± 10.5 μg/µL) is similar to 9*cis*-18:1 (320.7 ± 89.6 μg/µL), the latter being oleic acid, the most present MUFA in cells and cell membranes.

The analytical procedures for positional fatty acid isomer analysis were carried out using appropriate methodologies, i.e., dimethyl disulphide (DMDS) and iodine [43,44], as described in the Methods. In Appendix A, representative traces of GC and GC-MS analyses of DMDS adducts are reported (Appendix A: (A) GC analysis and (B) mass spectra of the DMDS adducts of fatty acid methyl esters obtained from membrane phospholipids in the fatty acid supplementation experiments). In this protocol we also determined the presence of geometrical *trans* isomers of the unsaturated fatty acids (Appendix A), in particular evidencing the *trans* isomers of palmitoleic and sapienic acids (9*trans*-16:1 and 6*trans*-16:1) as well as the *trans* isomer of oleic acid (9*trans*-18:1) and the total *trans* isomer content. It is not in the scope of this paper to focus on the follow-up of the *trans* isomer formation, however it is worth noting that at short times (up to 3 h monitoring) *trans* isomer levels were not significantly affected by the MUFA supplementation. After 24 h incubation with 300 µM sapienic acid the *trans* isomer content in membranes was significantly increased (Appendix A), whereas with 300 µM palmitic acid it was significantly increased in 1 h and then reduced after 24 h (Appendix A).

The high concentration used (300 µM) of the three fatty acids takes into account that in case of palmitoleic acids this is a critical concentration causing cell death, whereas for sapienic acid the cell viability remains at 25%. The fatty acid based-membrane phospholipid analysis up to 24 h is reported in Appendix A.

For comparison, Figure 4 shows the membrane phospholipid content build-up of sapienic acid concentration and of its *n*-10 metabolites, 8*cis*-18:1 and sebaleic acid (5*cis*,8*cis*-18:2), the *n*-6 arachidonic acid (20:4) and the *n*-3 22:6 docosahexaenoic acid (DHA) (as % of controls), for the supplementations of 150 µM (Figure 4A) and 300 µM (Figure 4B) sapienic acid as a function of time (0.5, 1, 3, 24 h) (Appendix A). At a high dosage of 300 µM sapienic acid it was interesting to observe the significant diminution of the *n*-3 DHA after 3 h (Figure 4B; see also Appendix A).

Since formation of lipid droplets was evidenced in the microscope images (Figure 2B), we treated the whole cell lipid extract of all experiments to evaluate the presence of triglycerides, separate the lipid classes and examine their fatty acid contents, as described in the Methods. Triglycerides were isolated in palmitoleic and palmitic acid supplementations after 24 h incubation at 150 and 300 µM (Appendix A). For sapienic acid supplementation, triglycerides were formed and isolated at all incubation times and at both concentrations used (150 and 300 µM in Appendix A, respectively). We were particularly interested in the behaviour of sapienic acid in this lipid class since this fatty acid is known to be associated with triglyceride composition in human sebum, as explained in the introduction [21], but no other information is available on its fatty acid metabolites. After 1 h an increase of sapienic acid in triglycerides up to 174.9 ± 1.5 µg/mL was detected and at 24 h this value was reduced to 62.1 ± 0.3 µg/mL (Appendix A). The two metabolites of sapienic acid, 8*cis*-18:1 and 5*cis*,8*cis*-18:2, were increasingly present in triglycerides along the incubation time, correspondently to their increases in membrane lipidome (see Figure 4A). It is worth noting that in all experiments we found oleic acid (9*cis*-18:1) to be the most representative MUFA in triglycerides.

Another interesting observation was that the lipid class of cholesteryl esters was formed only in the sapienic acid supplementation. This class was present at 1 and 3 h incubation and its full fatty acid composition was determined evidencing the presence of *n*-10 MUFA and PUFA fatty acids (Appendix A).

### 2.5. Generalized Polarization (GP) of Plasma Membrane in Caco-2 Cells

Information on the changes that the three fatty acids induce to the biophysical cell properties was obtained by generalized polarization (GP), a parameter yielding information on the fluidity of membranes, determined by using the Laurdan two-photon fluorescent microscopy [40,41]. Laurdan, 2-dimethylamino(6-lauroyl)naphthalene, is a fluorescent dye highly sensitive to the presence and mobility of water molecules within the membrane bilayer, yielding information on membrane fluidity by a shift in its emission spectrum [40]. Laurdan exhibits an emission spectral shift depending on the lipid phase state, i.e., bluish in ordered, gel phases and greenish in disordered, liquid–crystalline phases. Laurdan distributes equally between lipid phases and does not associate preferentially with specific fatty acids or phospholipid headgroups. In Figure 5, the results after 4 h incubation of the Caco-2 cells under 150 µM supplementation of palmitic, palmitoleic and sapienic acids are shown. In Figure 5A, representative Laurdan fluidity maps of Caco-2 cells are reported for untreated cells (Ctrl), and 150 µM of each fatty acid in a two-coloured pseudoscale, spanning from red (low GP, corresponding to a very fluid, disordered liquid crystalline state) to green (high GP, correspondent to a less fluid, ordered gel-like state). In all cases, high GP values were evident on membranes, which are generally less fluid than inner organelles in the cells [40,41,45]. Due to optimal resolution, in case of sapienic acid-treated cells it was evident from the pseudo-coloured fluidity scale that the cytoplasmic compartment was more fluid. The reason for such different distribution obtained with sapienic acid compared to palmitoleic acid (Figure 5A, bottom) is not straightforward, considering the little difference between these two structures, differing only by the two carbon atoms shift of the double bond position. The images also allow the plasma membrane, perinuclear and cytoplasmic organelles as well as lipid droplets to be examined in controls and other fatty acid treatments. These preliminary observations encourage more studies to be carried out using this powerful tool.

To detail the behavior of the fluidity distribution, GP histograms of the fluidity images are reported in Figure 5B, calculated as the normalized number of pixels with a particular value of fluidity (GP). The range of the histograms goes from 0 (very fluid regions) to 1 (very rigid regions). The histogram of untreated cells (Ctrl) is bimodal: There is a peak of a more fluid region (perinuclear and cytoplasmic) at 0.41, and another of a less fluid region (mainly plasma membranes) at 0.61. The mean value of the histogram is 0.44 (Figure 5C). The effect of palmitic acid on the membrane physical state is a transition to a more ordered, gel-like state, as noticeable in the histogram by the coalescence of the two peaks in a single peak at 0.63 (mean value 0.57, Figure 5C).

Palmitoleic acid showed a GP behavior intermediate between palmitic acid and sapienic acid, with a less pronounced transition to a less fluid, ordered state (mean value 0.57, Figure 5C). In the case of sapienic acid there was a much less marked transition, as evidenced from the smallest mean value variation respect to control cells, and from the less evident coalescence of the two peaks (mean value 0.48, Figure 5C, bottom graph). The data of the treated cells were statistically evaluated, as described in the Methods, and appear to be significant compared to controls (*p* < 0.05).

## 3. Discussion

Palmitic acid (16:0) is a common fatty acid present in the diet that does not have a negative effect on the viability of Caco-2 cell type [38,39], as also assayed in our conditions evaluating the EC_50_ (at 24 h, EC_50_ = 218.9 µM, see Table 1). It is interesting to note that only palmitic acid halved its EC_50_ to 100 µM in the time window 48–96 h, whereas for the C16 MUFAs the EC_50_ decreased 12–17% respect to the 24 h values (Table 1). By establishing the EC_50_ values, we proved that the two MUFA positional isomers are quite tolerated for 24 h at high concentrations (EC_50_ = 240.7 µM for palmitoleic acid and 262.2 µM for sapienic acid, see Table 1). These EC_50_ thresholds did not change significantly for long incubation times up to 96 h. These data confirmed also that 150 µM, although not certainly similar to “nutritional” doses, was not producing toxic effects in the time window of our observations, so that the metabolic changes did not express any evident cell impairment. The death pathways were also followed and the caspase activation, reported in Figure 3B, suggests that palmitic acid is able to trigger apoptosis as well as activate the necroptotic process. In case of palmitic acid we can interpret the low percentage of AnnV positivity (see Figure 3A) not as a lack of apoptotic cells, but as a consequence of the profound membrane alterations, due to the high incorporation of palmitic acid, as shown by the membrane lipid remodelling. This could prevent the flip-flop of phosphatidylserine residues to the outer face of the plasma membrane and the consequent binding of AnnV. Indeed, decreased fluidity parameters occurring during the cultivation with saturated fatty acid combinations in Caco-2 cell lines were previously examined [38]. In our previous studies using 150 µM palmitic acid supplementation in neuroblastoma cell lines [35] we were able to demonstrate that apoptosis was reverted either by washing the cell medium after 1 h supplementation (pulse and chase experiment) or by associating oleic and arachidonic acids to this saturated fatty acid in a mixed supplementation. Since a role for lipoapoptosis is proposed in the fatty acid supplementation [46], we could prove that, beside the lipid quantity, the quality of lipids is a strong driver of the cell fate. This is an important molecular basis to consider in lipid metabolic changes connected to favourable conditions for tumor onset or development, and indicates the need for investigations on lipid strategies, also using nutritional sources and deepening their influence on pharmacological treatments. The work presented here is addressed to palmitic acid biochemistry forming MUFAs, a fundamental step for cancer cells, also emerging for therapeutic applications [13,14,15,16,17,47]. Using the two MUFA positional isomers we could simulate the effects of the *n*-7 series with palmitoleic acid, for the prevalence of delta-9 desaturase, and of the *n*-10 series with sapienic acid for the prevalence of delta-6 desaturase, as depicted in scheme 1 of the Introduction. It is worth noting that these two MUFAs are not principally present in foods, therefore they are mainly connected to metabolism and are proposed as biomarkers of the partition of palmitic acid between the two desaturase enzymes, in the onset of cancer or other diseases and in the study of genetic polymorphisms of corresponding genes [27,48].

Palmitoleic and sapienic acids have a different fate in cells and at short times (0.5, 1, 3 h) distinct metabolism was evidenced (Table 2), with a very rapid elongation of sapienic acid to 8*cis*-18:1 and incorporation in membrane phospholipids, as well as in other lipid classes. Remarkably, the quantity of 8*cis*-18:1 in membrane phospholipids becomes practically similar to 9*cis*-18:1 (the main MUFA in human cell membranes of all tissues) after 24 h at 150 µM sapienic acid supplementation, indicating a previously unknown capability of the former to act as an important fatty acid component in cell membranes.

Moreover, for the first time in cancer cell metabolism it was also shown the rapid incorporation in membrane phospholipids of sebaleic acid, the *n*-10 PUFA obtained from the 8*cis*-18:1 desaturation (Figure 1, Table 2 and Figure 4) using delta-5 desaturase. The significant diminution of SFA (both 16:0 and 18:0, i.e., palmitic and stearic acids, respectively) at 3 and 24 h is another distinction from palmitoleic acid that can have influence on the membrane biophysical characteristics. In fact, the GP measurements after 4 h clearly showed the peculiar effects of sapienic acid compared to its positional isomer (Figure 5). The fluidity parameter was well differentiated between the two positional MUFA isomers with a diffuse effect of increased fluidity in all compartments in case of sapienic acid. The formation of *n*-10 MUFA, the PUFA sebaleic acid and the diminution of SFA can clearly contribute to the increase of fluidity observed in the cells.

The present investigation on sapienic acid supplementation expands knowledge on this hexadecenoic acid isomer, already known in bacteria [49,50,51], by addressing for the first time its fate in a popular human cancer cell model as Caco-2 cells. Since sapienic acid metabolism so far is prevalently connected to a component of human sebum [19,20,29,49], our findings suggest a previously unknown “systemic” effect due to its incorporation in cell lipidome. To our knowledge only one report describes the metabolism starting from sebaleic acid concerning human neutrophils with an interesting transformation to a chemoattractant by oxidative enzymatic pathways [52]. In a recent paper on phagocytic cells [28] some more information on sapienic acid was reported: In murine macrophages it showed anti-inflammatory activity at 25 µM concentrations, higher that those required for the same activity for *n*-7 and *n*-9 hexadecenoic MUFA; in murine RAW264.7 and P388D_1_ cell lines it was detected at comparable levels with the *n*-7 isomer and higher than the *n*-9 isomer; THP-1, a human cancer cell line, reported high levels of this *n*-10 MUFA. However, no follow-up of the sapienic acid metabolism to *n*-10 PUFA was reported. We underline that the de novo *n*-10 PUFA synthesis is a new aspect in cell biochemistry, which must be addressed for all types of eukaryotic cells, which are incapable of de novo synthesis of essential *n*-6 and *n*-3 PUFA [53]. Sebaleic acid can indeed be the first and the only de novo synthesized PUFA, as demonstrated here in human Caco-2 cells, in connection with the partition of palmitic acid between delta-6 and delta-9 desaturase enzymes, and with a new aspect of delta-5 desaturase activity on MUFA, possibly connected with genetic polymorphism [23,48].

Finally, it is worth underlining that the need of robust and standardized analytical protocols for lipidomics is well recognized [54], and sebaleic acid recognition was an unaddressed challenge with its mass exactly the same with linoleic acid (9*cis*,12*cis*-18:2). In fact, its structure with 18 carbon atoms chain and two double bonds separated by a methylenic carbon atom is an isomeric structure of the *n*-6 linoleic acid, having double bonds in C5 and C8 instead of C9 and C12. The GC analysis is a gold standard for resolving these two PUFA isomers (Appendix A). Remarkably, it was not noted in previous analyses of Caco-2 cell fatty acid supplementations [37,38,39]. We added the DMDS-adducts formation and corresponding mass spectrometry recognition of fragments (Appendix A) for the unambiguous assignment of MUFA and PUFA in our samples.

More work in other cell and disease models is in progress in order to thoroughly assess the full metabolic scenario with possible implications of the positional isomerism for health. These findings can be also extended to the studies of exosome composition in normal and tumoral cells [55].

## 4. Materials and Methods

Sapienic acid and the corresponding methyl ester, 8*cis*-18:1 methyl ester, and sebaleic acid methyl ester were purchased from Lipidox (Lidingö, Sweden); palmitic acid, palmitoleic acid, commercially available *cis* and *trans* FAME, dimethyl disulphide, iodine, RPMI 1640, Fetal Calf Serum (FCS), l-Glutamine, antibiotics, trypan blue and trypsin/EDTA, necroptosis inhibitor necrostatin-1 (Nec-1), dimethyl sulfoxide (DMSO) were purchased from Sigma-Aldrich (San Louis, MO, USA), and used without further purification; chloroform, methanol, diethyl ether and *n*-hexane, were purchased from Baker (HPLC grade) and used without further purification. Flasks and plates were from Falcon, BD Biosciences (Franklin Lakes, NJ, USA). CellTiter 96 Aqueous One Solution Cell Proliferation Assay and Caspase-Glo™ 3/7 luminescent assay were from Promega Corporation (Madison, WI, USA); the AnnexinV-EGFP Apoptosis Kit was purchased from BioVision, Inc. (Milpitas, CA, USA); the pan-caspase inhibitor carbobenzoxy-valyl-alanyl-aspartyl-[O-methyl]-fluoromethylketone (Z-VAD-fmk, hereinafter indicated as Z-VAD) was purchased from Vinci-Biochem (Florence, Italy). Laurdan, 2-dimethylamino(6-lauroyl)naphthalene (Laurdan, Molecular Probes, Inc., Eugene, OR, USA); other reagents used were from Carlo Erba, Milan, Italy.

Silica gel analytical and preparative thin-layer chromatography (TLC) was performed on Merck silica gel 60 plates, 0.25 mm thickness, and spots were detected by spraying the plate with cerium ammonium sulfate/ammonium molybdate reagent.

Fatty acid methyl esters (FAME) were analysed by GC (Agilent 6850, Milan, Italy) equipped with a 60 m × 0.25 mm × 0.25 µm (50%-cyanopropyl)-methylpolysiloxane column (DB23, Agilent, USA), and a flame ionization detector with the following oven program: Temperature started from 165 °C, held for 3 min, followed by an increase of 1 °C/min up to 195 °C, held for 40 min, followed by a second increase of 10 °C/min up to 240 °C, and held for 10 min. A constant pressure mode (29 psi) was chosen with helium as carrier gas. Methyl esters were identified by comparison with the retention times of authentic samples. Calibration procedures for quantitative analyses followed reported procedures [21], using a C17:0 fatty acid as internal standard, whereas for fatty acids such as 8*cis*-18:1 and 5*cis*,8*cis*-18:2 data are given in the S1 protocol reported in Appendix A. The data are given as µg/mL of the FAME identified in three independent experiments, and values are reported as mean ± SEM (standard error of the mean).

Dimethyl disulphide adducts of FAME were analysed by GC-MS (Thermo Scientific Trace 1300, Waltham, MA, USA) equipped with a 15 m × 0.25 mm × 0.25 µm TG-SQC 5% phenyl methyl polysiloxane column, with helium as carrier gas, coupled to a mass selective detector (Thermo Scientific ISQ) with the following oven program: Temperature started at 80 °C, maintained for 2 min, increased at a rate of 15 °C/min up to 140 °C, increased at a rate of 5 °C/min up to 280 °C and held for 10 min.

Viability was evaluated by measuring absorbance at 490 nm by a microtiter plate reader Multiskan EX (ThermoLabSystems, Basingstoke, UK). Phase contrast microscopy was carried out with a Nikon Eclipse TS100 (Tokyo, Japan) microscope equipped with a digital camera. Flow cytometry analysis was performed on a FACSAria BD Analyser using FACSDiva software (Becton, Dickinson and Company, Franklin Lakes, NJ, USA).

Laurdan intensity images were obtained with an inverted confocal microscope (SP2, Leica Microsystems, Wetzlar, Germany) using a 63× oil immersion objective (NA 1.4) under excitation at 800 nm with a mode-locked Titanium-Sapphire laser (Chamaleon, Coherent, Santa Clara, CA, USA). Internal photon multiplier tubes collected images in an eight bit, unsigned images at a 400 Hz scan speed. Laurdan intensity images were recorded simultaneously with emission in the range of 400–460 nm and 470–530 nm and imaging was performed at room temperature. A stack consisting of 10 z-sections every 2 µM was acquired for each field of view. Frequency represents the normalized number of pixels per cell having a specific GP value. The analysis was performed on n = 80 cells per sample before and after 4 h of the specific treatments.

### 4.1. Cell Cultures

Caco-2 cells (ATCC^®^ Number: HTB-37™), derived from a human colorectal adenocarcinoma were from the departmental cell collection. Cells were cultured at 37 °C in humidified atmosphere at 5% CO_2_ in RPMI 1640 supplemented with 10% heat-inactivated FCS, 2 mM l-glutamine, 100 units/mL penicillin, 0.1 mg/mL streptomycin (hereafter referred to as complete medium). Cultures were maintained in the log phase of growth with a viability >95%. Cells were checked for the absence of mycoplasma infection.

### 4.2. Cell Viability Evaluation

Caco-2 cells (2 × 10^3^ cells/well) were seeded onto 96-well microtiter plates in 100 µL of complete medium. After 24 h, the medium was removed from each well and the fatty acids, dissolved in ethanol, adjusted at the final concentration, ranging from 100 to 300 µM in complete medium, were immediately added to cells (final concentration of ethanol <1%). Control samples were carried out in presence of ethanol at the same concentration in free fatty acid samples. Moreover, no variations in cell viability were observed in control cells incubated in the presence or in the absence of ethanol. Viability was determined after the indicated times by adding 20 µL/well of CellTiter 96 Aqueous One Solution Cell Proliferation Assay, as described in [56]. The absorbance at 490 nm was measured after 1 h incubation at 37 °C by a microtiter plate reader Multiskan EX (ThermoLabSystems). Cell viability was expressed as percentage of control values obtained from cultures grown in the absence of fatty acids.

Time-course experiments were performed using the first concentration of each fatty acid causing a cell viability reduction ≥75% after 48 h from the fatty acid supplementation (i.e., 150 µM palmitic acid, 250 µM palmitoleic acid and 300 µM sapienic acid). Cells were exposed to fatty acids for times ranging from 2 to 96 h. The EC_50_ (fatty acid effective concentration to reduce cell viability by 50% after 48 h) and ET_50_ (time required to reduce cell viability by 50%) were calculated using linear regression analysis.

Cell viability was also evaluated at 24 h in Caco-2 cells pretreated with Z-VAD (100 µM) or Nec-1 (100 µM), added to cells 3 h before fatty acid supplementation. AnnexinV and Propidium iodide positivity were detected through flow cytometry, using the AnnexinV-EGFP Apoptosis Kit, as described in [57]. The cells (2 × 10^5^/3 mL) were seeded in 25 cm^2^ flasks and, after 24 h, treated with fatty acids. After treatment, the cells were centrifuged at 500× *g* for 5 min, washed in 2 mL complete medium, centrifuged again and stained according to the manufacturer’s instructions. Within 30 min, the cells were analysed through flow cytometry on a FACSAria BD Analyser using FACSDiva software version 4.1, 2004 (BD Biosciences, San Jose, CA, USA). The morphological analysis of the treated cells was conducted through phase contrast microscopy directly in 96-well plates using a Nikon Eclipse TS100 microscope.

### 4.3. Caspase Activation

The activity of caspase 3/7 was assessed by the Caspase-Glo™ 3/7 luminescent assay. The cells (2 × 10^3^/well) were seeded onto 96-well microtiter plates in 100 μL of complete medium. After 24 h, cells were supplemented with 150 µM palmitic acid, 250 µM palmitoleic acid and 300 µM sapienic acid. After incubation for the indicated time, the medium was removed from each well and replaced with 50 µL of complete medium and 50 µL of Caspase-Glo™ 3/7 [58]. Luminescence was measured by Fluoroskan Ascent FL (Labsystem, Helsinki, Finland) according to the manufacturer’s instructions. Luminescence values were normalised to cell viability.

### 4.4. Western Blot Analysis of p38 and cPLA_2_

Cells (3×10^6^/20 mL) were seeded in 75 cm^2^ flasks and, after 24 h, the medium was supplemented with 150 or 300 µM palmitic acid, 250 µM palmitoleic acid or 300 µM sapienic acid. After 180 min, cells were harvested with a cell scraper, collected by centrifugation at 500× *g* for 5 min and washed twice in PBS. Cell pellets were lysed with 100 µL of Cell Lytic-M supplemented with the Protease Inhibitor Cocktail (1:100), Phosphatase Inhibitor Cocktail 1 (1:100) and sodium-orthovanadate (1:500) (Sigma-Aldrich). After 45 min at 0 °C and vortexing every 5 min, insoluble material (nuclear pellet plus membranes) was removed by centrifugation at 12,000× *g* for 30 min at 4 °C. Protein supernatant (cell lysate) was collected and stored at −80 °C. Protein content was quantified by spectrophotometer using Bradford’s method (Bio-Rad Protein Assay, Bio-Rad, Hercules, CA, USA) and 80 µg/lane of protein were separated by SDS-PAGE (10% gel) and blotted to nitrocellulose membrane (150 V for 90 min). Non-specific antibody binding sites were blocked by incubation with blocking buffer, TRIS buffered saline, 0.1% Tween 20 (TBS/T) with 5% *w*/*v* non-fat dry milk, for 1 h at room temperature. After five washes with TBS/T, membranes were incubated overnight at 4 °C with anti-phospho-cPLA_2_ (Ser 505) or phospho-p38 mAbs (Cell Signaling Technology, Inc.; Beverly, MA, USA) diluted in TBS/T with 5% bovine serum albumin, according to the manufacturer’s instructions. After a further five washes with TBS/T, membranes were incubated for 1 h at room temperature with horseradish peroxidase–conjugated anti-rabbit antibody (Sigma-Aldrich) diluted in blocking buffer. After a further five washes, proteins were detected by incubating the membrane with Immobilon Western detection reagent (Millipore, Burlington, MA, USA). The anti-phospho antibodies were then stripped for 30 min in 25 mM glycine-HCl pH 2, 1% SDS (*w*/*v*), and, after blocking with non-fat milk, the membrane was re-incubated with antibodies recognising total cPLA_2_ or total p38 (Cell Signaling) to account for equal loading. Proteins were detected as above. The relative levels of expression of different proteins were determined using ImageJ software [35].

### 4.5. Lipid Extraction and Fatty Acid Analysis

A constant amount of cells (counting 6×10^6^ millions of cells) in a 1.5 mL vial was added with tridistilled water (2 mL) and 2:1 chloroform/methanol (4 times × 4 mL of 2:1 chloroform/methanol mixture) to extract lipids according to the Folch method [59]. The organic layers, dried on anhydrous Na_2_SO_4_ evaporated to dryness, gave the total lipid extract that was checked by thin layer chromatography for lipid class separation as previously reported [21]. In some of the cell samples triglycerides and cholesteryl esters were also obtained, therefore a chromatographic separation of the lipid classes was performed and differentiated conversion of the lipid classes was performed: Fatty acid-containing phospholipids were transformed to the corresponding FAME by adding 0.5 M solution of KOH in MeOH (0.5 mL), quenching the reaction after 10 min for PL fraction and 30 min for TG fraction by brine addition (0.5 mL). The derivatization of fatty acid moieties in cholesteryl esters was carried out by adding 1 M solution of NaOH in 3:2 MeOH/benzene (0.5 mL). The reaction was stirred in the dark under argon and quenched by brine (0.5 mL) after 15 min. FAME were extracted with *n*-hexane (3 times × 2 mL of *n*-hexane), dried on anhydrous Na_2_SO_4_, evaporated to dryness and analysed by GC in comparison with standard references. Detailed fatty acid compositions of each lipid class identified in the experiments with 150 and 300 µM palmitic, palmitoleic and sapienic acids supplementations are listed as µg/mL in Appendix A.

### 4.6. DMDS Derivatization

The FAME mixture obtained from the Caco-2 cells supplemented with sapienic acid was treated following a previously described procedure for the assignment of the double bond position [43,44]. Briefly, in a Wheaton vial containing FAME in *n*-hexane (50 μL), 100 μL of dimethyl disulfide and 2 drops of a 6% solution of iodine in diethyl ether were added. The reaction was stirred at room temperature for 30 min, under argon atmosphere. Then 1 mL of *n*-hexane and 1 mL of a 5% aqueous solution of sodium thiosulphate were consecutively added. The organic phase was isolated, dried over anhydrous Na_2_SO_4_, and concentrated under gentle stream of nitrogen, before the GC-MS analysis (see Appendix A).

### 4.7. Laurdan Two-Photon Microscopy

Cells were seeded in uncoated Petri dish (cat. #80136, IbiDi, Martinsried, Germany) at 10^5^ cells/well. After 48 h equilibration in cell incubator, cells were labeled with 1 µL of Laurdan (2-dimethylamino(6-lauroyl)naphthalene) stock solution per milliliter of RPMI 1640. The stock solution concentration was 1 mM in dimethyl sulfoxide (DMSO) and was renewed every three weeks. After 30 min equilibration in the cell incubator, the medium was supplemented with 150 µM palmitic acid, 150 µM palmitoleic acid, and 150 µM sapienic acid. Imaging was performed before treatment and after 4 h. As a normalized ratio of the intensity at the two emission wavelengths regions, the generalized polarization (GP) provides a measure of membrane order, in the range between −1 (liquid-crystalline) and +1 (gel). The GP, defined as
(1)$$GP=\frac{I_{\left(400–460\right)}-GI_{\left(470–530\right)}}{I_{\left(400–600\right)}+GI_{\left(470–530\right)}}$$
was calculated for each pixel using the two Laurdan intensity images (*I*_(400–460)_ and *I*_(470–530)_) by using the program Ratiometric Image processor [43,44]. The calibration factor *G* was obtained from the GP values of solutions of Laurdan in DMSO. *G* factor had approximately 2% variation across the imaging area. GP images (as eight-bit unsigned images) were pseudo-coloured in ImageJ software (current version 1.48V, Java 1.6.0_65). Background values (defined as intensities below 7% of the maximum intensity) were set to zero.

### 4.8. Statistical Analysis

Results by cell experiments are given as mean values ± SD. Statistical analyses of data from cell viability, caspase experiments and western blot analysis were conducted using the XLSTAT-Pro software, version 6.1.9 (Addinsoft, 2003, Brooklyn, NY, USA) with a 95% confidence interval (*p* ≤ 0.05). Data were analysed by ANOVA and ANCOVA with Bonferroni’s correction, followed by a comparison with Dunnett’s test. Analysis of acquired images for the microscopy experiments were performed with ImageJ. For GP data, mean ± SD values were determined and utilized for two-tailed Student’s *t*-test analysis. GP were averaged over n = 80 cells per sample. Fatty acids results are given as mean ± SEM (standard error of the mean). Statistical analysis was performed using GraphPad Prism 5.0 software (GraphPad Software, Inc., San Diego, CA, USA). We used non-parametric unpaired *t*-test two-tailed with 95% confidence interval.

## 5. Conclusions

Our work offers the first evidence that sapienic and palmitoleic acids are two positional hexadecenoic fatty acid isomers with a distinct metabolism in human Caco-2 cells, the former being rapidly converted to *n*-10 C18 MUFA and a *n*-10 PUFA, increasing the unsaturated fatty acid levels in the cell lipidome, contributing to an increased fluidity and improving cell survival. These aspects must be investigated in detail in different cancer cells and conditions. The results here shown for Caco-2 cells open novel perspectives for lipidomic studies in health and diseases.

## Figures and Tables

**Figure 1 ijms-20-00832-f001:**
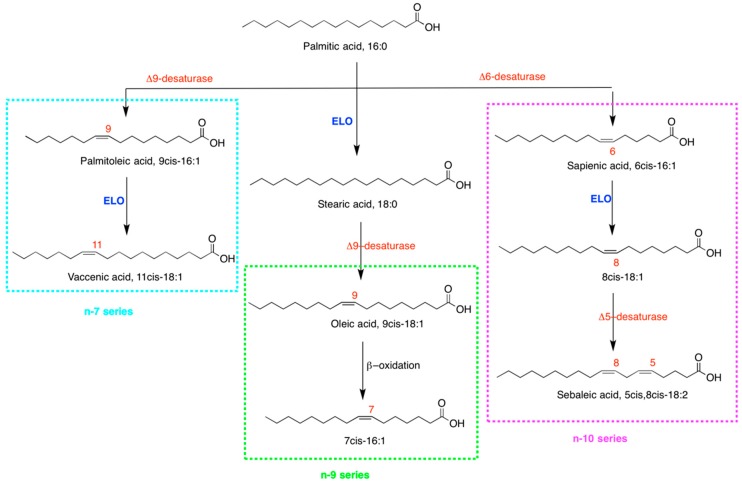
Biosynthetic access to three fatty acid series starting from palmitic acid: (**1**) *n*-7 series (blue box) formed via a first direct desaturation step to palmitoleic acid (9*cis*-16:1) by delta-9 desaturase (stearoyl-CoA desaturase, SCD1) creating a double bond in C9–C10 position, followed by elongation to vaccenic acid (11*cis*-18:1); (**2**) *n*-9 series (green box) with oleic acid (9*cis*-18:1) formed in two steps, including elongase enzymatic activity from palmitic to stearic acid (18:0) and subsequent desaturation creating the double bond in C9–C10 of the 18 carbon atom fatty acid chain; the beta oxidation step from oleic acid also brings to another monounsaturated fatty acid (MUFA) of the *n*-9 series, i.e., 7*cis*-16:1; (**3**) *n*-10 series (purple box) with the formation of sapienic acid (6*cis*-16:1) by delta-6 desaturase activity on palmitic acid followed by the elongation step to 8*cis*-18:1. The subsequent fate of 8*cis*-C18:1 is the transformation into a *n*-10 polyunsaturated fatty acid, 5*cis*,8*cis*-18:2 (sebaleic acid) via delta-5 desaturase enzyme.

**Figure 2 ijms-20-00832-f002:**
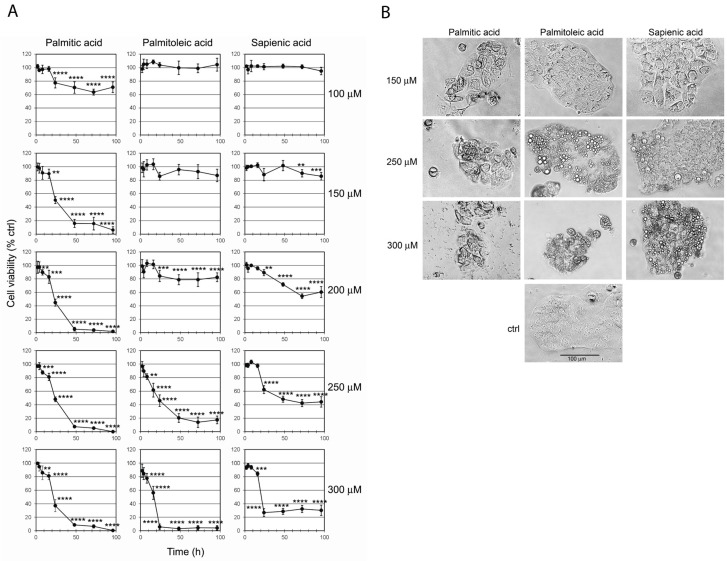
(**A**) Effect of fatty acid supplementation on Caco-2 cell viability expressed as relative percentages compared to control cells without supplementation. Cell viability was evaluated by a colorimetric assay based on MTS reduction. Cells were exposed for different times to the indicated concentrations of palmitic, palmitoleic or sapienic acids. Results are means ± SD of three different experiments, expressing the percentage of viability compared to control cultures. Values of SD never exceeded 15%. Data were analysed by an ANOVA/Bonferroni test, followed by a comparison with Dunnett’s test (confidence range 95%; * *p* < 0.05, ** *p* < 0.01, *** *p* < 0.001, **** *p* ≤ 0.0001 versus untreated cells). (**B**) Appearance of Caco-2 cells supplemented with different fatty acid concentrations for 24 h. Cell morphology was assessed by phase contrast microscopy after the exposure to the indicated concentrations of the three fatty acids. The cell morphology of control cells is also shown. Magnification 200×.

**Figure 3 ijms-20-00832-f003:**
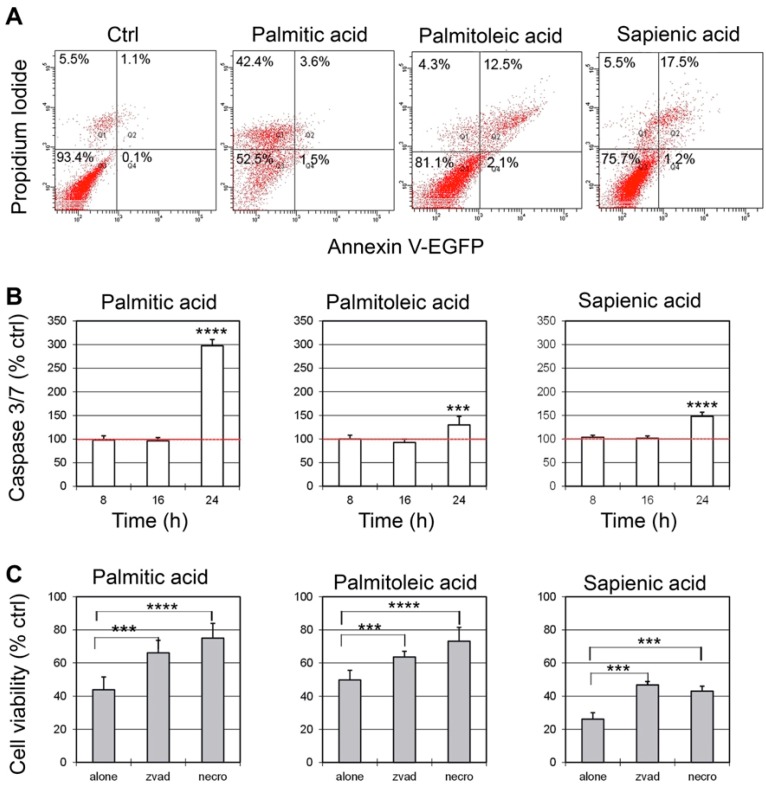
Evaluation of death pathways triggered by fatty acids on Caco-2 cells. (**A**) The presence of apoptotic/necrotic cells was investigated after AnnexinV/Propidium Iodide (PI) staining and flow cytometry analysis. Representative plots of AnnexinV (X-axis/PI (Y-axis) staining of Caco-2 cells are shown. (**B**) Caspase 3/7 activation was evaluated by a luminometric assay. Cells were exposed for 8, 16 or 24 h to 150 µM palmitic acid, 250 µM palmitoleic acid and 300 µM sapienic acid (at these concentrations 75–80% reduction of cell viability is observed). (**C**) Caco-2 cells were pretreated for 3 h with the pan-caspase inhibitor Z-VAD or with the necroptosis inhibitor necrostatin-1 (Nec-1) and then treated with 150 µM palmitic acid, 250 µM palmitoleic acid and 300 µM sapienic acid. Cell viability was measured after 48 h. Results are means of three different experiments each performed in triplicate and they represent the percentage of control values obtained from cultures grown in the absence of fatty acid. SD never exceeded 15%. (*** *p* < 0.001; **** *p* < 0.0001).

**Figure 4 ijms-20-00832-f004:**
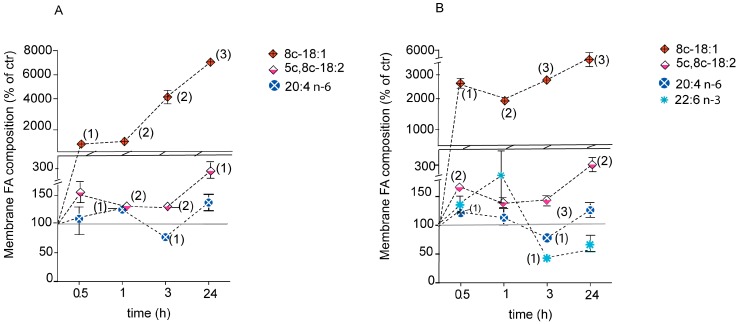
Time monitoring of significant MUFA and PUFA changes in membrane fatty acids of Caco-2 cells (reported as % of controls) treated with sapienic acid. (**A**) 150 µM and (**B**) 300 µM. The corresponding quantitative values are reported in Appendix A. The grey bar indicates the values of control cells. Numbers in parenthesis indicate the statistical significance value: (1) *p* ≤ 0.047; (2) *p* ≤ 0.0097; (3) *p* ≤ 0.0009 for (**A**); (1) *p* ≤ 0.0485, (2) *p* ≤ 0.0073, (3) *p* ≤ 0.0009 for (**B**).

**Figure 5 ijms-20-00832-f005:**
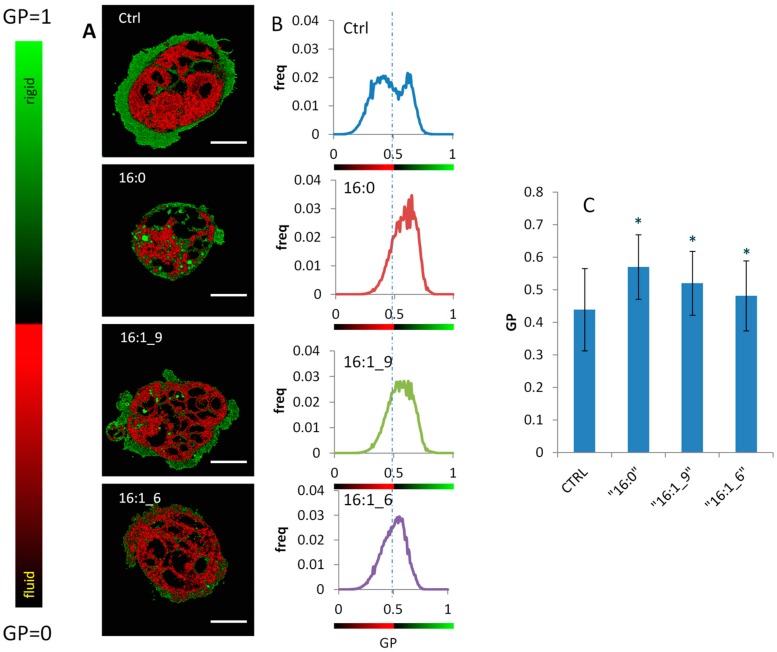
Membrane fluidity of Caco-2 cells resulting from different fatty acid supplementations. (**A**) Ratiometric Laurdan fluidity maps (scale bar 20 μm) of representative Caco-2 cells reported after 4 h for untreated cells (Ctrl), palmitic acid (16:0), palmitoleic acid (9*cis*-16:1) and sapienic acid (6*cis*-C16:1) 150 μM treatments in a two-coloured pseudoscale, spanning from red (very fluid) to green (very rigid). (**B**) GP histograms of the fluidity images for (from top to bottom): Untreated cells (Ctrl), palmitic acid (16:0), palmitoleic acid (9*cis*-C16:1) and sapienic acid (6*cis*-C16:1). The range of the histograms goes from 0 (very fluid regions) to 1 (very rigid regions). (**C**) Mean Values of GP histograms for the different treatments (* *p* < 0.05 compared to Ctrl).

**Table 1 ijms-20-00832-t001:** Fatty acid EC_50_ (μM) estimated on Caco-2 cell viability after the indicated incubation times. EC_50_ is the concentration of fatty acid required to reduce Caco-2 cell viability by 50%, calculated by linear regression. Viability was evaluated measuring tetrazolium salt reduction.

Fatty Acid	24 h	48 h	72 h	96 h
Palmitic acid	218.9	105.5	99.6	101.1
Palmitoleic acid	240.7	217.0	214.3	200.5
Sapienic acid	262.1	245.3	230.2	232.3

**Table 2 ijms-20-00832-t002:** Statistically significant trends of fatty acids in membrane phospholipids (increase ↑ or decrease ↓) after treatment with 150 µM PO (palmitoleic acid), SA (sapienic acid) and PA (palmitic acid) in the interval time of 0.5–3 h. The analysis of fatty acid as methyl esters (FAME) was carried out as reported in Section 2. Values are presented in Appendix A. Significance: * *p* ≤ 0.049, ** *p* ≤ 0.009; *** *p* ≤ 0.0001.

Fatty Acid	PO 0.5 h	SA 0.5 h	PA 0.5 h	PO 1 h	SA 1 h	PA 1 h	PO 3 h	SA 3 h	PA 3 h
16:0	↑ *				↓ *			↓ *	↑ *
*6cis*-16:1					↑ ***		↓ *	↑ ***	
*9cis*-16:1	↑ **				↓ **		↑ ***		
18:0				↓ *				↓ *	↓ *
*8cis*-18:1		↑ *		↓ *	↑ **			↑ **	
*5cis*,*8cis*-18:2					↑ **		↓ **	↑ **	↓ **
18:2 *n*-6								↓ *	↓ *
20:0	↑ **	↑ ***			↑ *	↑ *			
20:3 *n*-6				↓ *			↓ ***		
20:4 *n*-6				↓ *	↑ *		↓ *	↓ *	
20:5 *n*-3									
22:5 *n*-3				↓ **			↓ *		
22:6 *n*-3							↓ *		
SFA					↓ *			↓ *	
MUFA					↑ *		↑ *	↑ **	
Total PUFA				↓ *			↓ *		
PUFA *n*-6						↓ *	↓ ***	↓ *	↓ *
PUFA *n*-3				↓ *			↓ *

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
