# Peer review of "Hexadecenoic Fatty Acid Positional Isomers and De Novo PUFA Synthesis in Colon Cancer Cells"

_ijms, 2019, doi:10.3390/ijms20040832_

Reviewer 1 Report

The manuscript entitled “Hexadecenoic fatty acids and de novo PUFA synthesis in colon cancer cells”, aimed to show membrane fatty acids remodelling in Caco-2 cells treated with palmitoleic (9cis-16:1, n-7 series) and sapienic (6cis-16:1, n-10 series) acids. Lipid profiles of fatty acid-treated cells are correlated with specific cell signaling pathways and biophysical membrane properties. The authors provide evidence that n-7 and n-10 hexadecenoic fatty acid isomers have different fates in Caco-2 cells thus providing different PUFA species that are incorporated in membrane phospholipids. These data “highlight the importance of positional isomerism, showing that cellular fate and properties depended not only from the length of the carbon atom chain and the number of double bonds, but also from the position of the unsaturation along the chain, influencing biophysical and metabolic outcomes”.

This manuscript reports new data that add new information on the metabolism and biological role of MUFA and PUFA related to the geometrical and positional isomerism. The experimental design is well thought out and the conclusions drawn are consistent with the result reported. For these reasons the reviewer considers that this manuscript is well-suited for publication in the International Journal of Molecular Sciences. The reviewer does not have any major concerns except the following few suggestions.

Comments

a)     The English should be revised. Some paragraphs are too long and this makes the reading difficult.

b)    The text of the abstract should be modified in order to highlight the aim and the novelty of the study.

c)     Materials and Methods

-       the reviewer suggests modifying the organization of Materials and Methods such as 4.2 Cell cultures; 4.3 Cell viability evaluation; 4.4 Caspase activation; 4.5 Western blot analysis of p38 and cPLA2; 4.6 Lipid extraction and fatty acids analysis; 4.7 DMDS derivatization….

-       Please, change the title of paragraph 4.4 (suggested paragraph 4.2 of revised version) in Cell viability evaluation.

-       Since fatty acids are dissolved in ethanol the authors should specify whether the controls are cells incubated with ethanol (vehicle control).

-       Western blot analysis of p38 and cPLA2 (pag 15, line 621). Please, add the method used to quantify protein content.

-       Lipid extraction and Fatty acid analysis (pag.14, line 549 and line 560). The amount of 2:1 chloroform/methanol added to cells it is not clear as it is expressed as 4x4ml. Please, express this amount in vol.

d)    Results

-       Pag 4, line 139. Please, specify the EC50 relative to palmitoleic and sapienic acid.

-       Fig. 2: Please, add the label on y-axis.

-       Please, add one or more reference that report the activation of cPLA2 by p-p38.

-       Fig. 3: Please, report in the legend the concentration of fatty acids used (pag. 6, line 219).

e)     Discussion

Pag 12, lines 425-429: Authors should amend this sentence since it is not clear to which result it is referred to relate the activation of cPLA2 and the cell fate of palmitoleic and sapienic acids.

There are also some minor corrections to be made.

a)     Pag 2 line 62: Please, add the name of the enzyme before the abbreviation (SCD1)

Author Response

This reviewer did not have any major concerns except the few suggestions answered below. We thank this reviwer for his/her careful reading.

Comments

a)     The English should be revised. Some paragraphs are too long and this makes the reading difficult.

We amended the English  language by a native speaking person and shortened the lenght of some paragraphs as requested.

b)    The text of the abstract should be modified in order to highlight the aim and the novelty of the study

The abstract was modified as requested.

c)     Materials and Methods

-       the reviewer suggests modifying the organization of Materials and Methods such as 4.2 Cell cultures; 4.3 Cell viability evaluation; 4.4 Caspase activation; 4.5 Western blot analysis of p38 and cPLA2; 4.6 Lipid extraction and fatty acids analysis; 4.7 DMDS derivatization….

We followed the suggestion.

-           Please, change the title of paragraph 4.4 (suggested paragraph 4.2 of revised version) in Cell viability evaluation.

We followed this suggestion.

-       Since fatty acids are dissolved in ethanol the authors should specify whether the controls are cells incubated with ethanol (vehicle control).

The conditions of control cells were detailed in Methods page 14, lines 546-547. Control cells  were incubated with/without ethanol and no differences were found.

-       Western blot analysis of p38 and cPLA2 (pag 15, line 621). Please, add the method used to quantify protein content.

The quantification of protein content was detailed in the Methods section, page 15, lines 587-588

-       Lipid extraction and Fatty acid analysis (pag.14, line 549 and line 560). The amount of 2:1 chloroform/methanol added to cells it is not clear as it is expressed as 4x4ml. Please, express this amount in vol.

The indication was ameliorated: (page 15, line 606: 4 times × 4mL of 2:1 chloroform/methanol mixture; page 16, line 617: 3 times × 2 mL of n-hexane)

d)    Results

-       Page 4, line 139. Please, specify the EC50 relative to palmitoleic and sapienic acid.

As all the EC50 values are reported in Table 1, we retain to simplify the sentence at line 139 as follow: “The EC50s calculated after 24 hours for the three fatty acids were in the same concentration range (see Table 1)”. In page 4, lines 149-150 we detailed the concentrations in the brackets.

-       Fig. 2: Please, add the label on y-axis.

A revised Figure 2 is provided

-       Please, add one or more reference that report the activation of cPLA2 by p-p38

A new reference 40 was added and the reference numbering was amended accordingly.

-       Fig. 3: Please, report in the legend the concentration of fatty acids used (pag. 6, line 219).

DONE

e)     Discussion

Pag 12, lines 425-429: Authors should amend this sentence since it is not clear to which result it is referred to relate the activation of cPLA2 and the cell fate of palmitoleic and sapienic acids. 

The discussion was simplified and this sentence on the activation of cPLA2 was eliminated (page 12 line 447)

There are also some minor corrections to be made. 

a)     Pag 2 line 62: Please, add the name of the enzyme before the abbreviation (SCD1)

DONE

Reviewer 2 Report

In this manuscript, Roberta Scanferlato and co-authors discussed the role of hexadecenoic fatty acids and de novo PUFA synthesis in colon cancer cells. The study has some novelty since this topic hasn’t been discussed in the colon cancer before. However, the authors failed to make a clear conclusion based on their results. Instead of descriptive narration of how the experiments were designed and performed, the authors should organize more explicit points about what to set out in the paper, especially for abstract and title. The clear conclusions are necessary to raise the reader’s interest and very helpful for the reader to understand the structure and rationale of the paper. For example, in the abstract, the authors mentioned: "n-7 and n-10 MUFA supplementations induce different biophysical effects." The author should clarify which kind of biophysical effect. In addition, the different biophysical effects suggested by authors are not fully supported by the results. In Figure 5C, there is no statistical difference between palmitoleic acid (16:1_9) and sapienic acid (16:1_6). In Figure 2 and 3, the authors didn’t compare Palmitioleic acid and Sapienic acid in term of cell viability. The writing of this manuscript needs significant improvement to meet the criteria of publication. Several other concerns are listed as follows:

The label and annotation are hard to be recognized in Fig2. Please increase the size of image and label.

Small lipid droplets are hard to be identified by the bright field of a microscope. Staining cell with Nile red or Oil O Red will be more sensitive to monitor the lipid droplet formation, especially considering authors claimed palmitic acid did not show the presence of cytoplasmic droplets

Some parts of the writing are tedious and hard to follow. For instance, in result 2.2, the authors don’t need to repeat the narration in a cited study.  “These morphological changes in Caco-2 cells treated with high palmitic acid concentrations were previously described by van Greevenbroek [39] which, after electron microscopy analysis, observed that: “the palmitic acid-incubated cells were laden with membranes extending throughout the cells without accumulation of lipid droplets or lipoprotein-like particles in any distinct part of the cells”. " It looks more concise if authors delete the words after “which”.

In the method part, the authors mentioned general polarization (GP) provides a measure of membrane order, in the range between -1 (liquid-crystalline) 645 and +1 (gel). However, in figure 5 and result, scale bars were labeled as 0 to 1. The range of the histograms goes from 0 (very fluid regions) to 1 (very rigid regions). The authors should explain more about how 5B, C was prepared, how many cells were used for calculation, what is the unit of Frequency?

It seems that there is a shift pattern when compare 16:1_9 vs 16:1_6. However, the authors didn’t provide any statistical evidence prove this difference. Is there any other statistical analysis can show a significant difference between the effect of palmitoleic acid and sapienic acid?

The authors suggested palmitoleic acid and sapienic acid treatment induced the biological difference. Whether these differences in lipid metabolism process lead to the biophysical differences in colon cancer cell lines? In what mechanism?

FFA should be conjugate with BSA first when being added into the medium, did authors treat the cells in this way?  It should be described in the method part.

There are several careless mistakes in this manuscript. For example, Figure 4C should be 5C in result 2.5.

Author Response

In this manuscript, Roberta Scanferlato and co-authors discussed the role of hexadecenoic fatty acids and de novo PUFA synthesis in colon cancer cells. The study has some novelty since this topic hasn’t been discussed in the colon cancer before. However, the authors failed to make a clear conclusion based on their results. Instead of descriptive narration of how the experiments were designed and performed, the authors should organize more explicit points about what to set out in the paper, especially for abstract and title.

We thank this reviewer for the critical reading and his/her suggestions that we followed in the revised version.

The clear conclusions are necessary to raise the reader’s interest and very helpful for the reader to understand the structure and rationale of the paper. For example, in the abstract, the authors mentioned: "n-7 and n-10 MUFA supplementations induce different biophysical effects." The author should clarify which kind of biophysical effect. In addition, the different biophysical effects suggested by authors are not fully supported by the results. In Figure 5C, there is no statistical difference between palmitoleic acid (16:1_9) and sapienic acid (16:1_6).

 The biophysical effect was better explained in the abstract (page 1, lines 30-31) and in the Discussion (page 12, lines 447-454): “The fluidity parameter was well differentiated between the two positional MUFA isomers with a diffuse effect of increased fluidity in all compartments in case of sapienic acid. The formation of n-10 MUFA, the PUFA sebaleic acid and the diminution of SFA can clearly contribute to the increase of fluidity observed in the cells.” This metabolism derived from sapienic acid and not from palmitoleic acid.

In Figure 2 and 3, the authors didn’t compare Palmitioleic acid and Sapienic acid in term of cell viability.

The cell viability is discussed in the manuscript in section 2.1, page 4, with the EC50 values (Table 1) and the cell viability monitoring wasreported by comparing the three fatty acids at different concentrations and times (Figure 2 panel A).

The writing of this manuscript needs significant improvement to meet the criteria of publication. Several other concerns are listed as follows:

 We rewrote several sections and shortened some points in order to improve the reading.

The label and annotation are hard to be recognized in Fig2. Please increase the size of image and label.

DONE

Small lipid droplets are hard to be identified by the bright field of a microscope. Staining cell with Nile red or Oil O Red will be more sensitive to monitor the lipid droplet formation, especially considering authors claimed palmitic acid did not show the presence of cytoplasmic droplets

The images reported in Figure 2 Panel B are indeed not useful to monitor lipid droplets. In fact we used the chromatographic isolation of triglycerides, with quantification of the fatty acid species, as reported on page 10 from line 339. We  better expressed this strategy, which overcomes the use of imaging tools.

Some parts of the writing are tedious and hard to follow. For instance, in result 2.2, the authors don’t need to repeat the narration in a cited study.  “These morphological changes in Caco-2 cells treated with high palmitic acid concentrations were previously described by van Greevenbroek [39] which, after electron microscopy analysis, observed that: “the palmitic acid-incubated cells were laden with membranes extending throughout the cells without accumulation of lipid droplets or lipoprotein-like particles in any distinct part of the cells”. " It looks more concise if authors delete the words after “which”.

DONE

In the method part, the authors mentioned general polarization (GP) provides a measure of membrane order, in the range between -1 (liquid-crystalline) 645 and +1 (gel). However, in figure 5 and result, scale bars were labeled as 0 to 1. The range of the histograms goes from 0 (very fluid regions) to 1 (very rigid regions).

The range [-1;1] is the measurement field of the instrument. In general, the distribution of GP values of cells is characterized by values ranging in a narrower interval [0-1]. It is like a thermometer that can measure -100 to +40 °C, but when used to measure ambient temperature it never reaches -100 °C. In the measurements only the “non-zero” range of the histograms is in fact reported.

The authors should explain more about how 5B, C was prepared, how many cells were used for calculation, what is the unit of Frequency?

We detailed in the Methods (page 14, line 532) that the “Frequency represents the normalized number of pixels per cell having a specific GP value”. In the methods (page 14 line 533) and in statistics section (page 16 line 654) we added the number of cells (n=80 per sample).

For clarity, we wrote on Page 10 line 379: “In Figure 5B, GP histograms of the fluidity images are reported, calculated as the normalized number frequency of pixels with a particular value of fluidity (GP)

It seems that there is a shift pattern when compare 16:1_9 vs 16:1_6. However, the authors didn’t provide any statistical evidence prove this difference. Is there any other statistical analysis can show a significant difference between the effect of palmitoleic acid and sapienic acid?

In the first version, in the section of Statistical analysis (page 16, line 654) it was reported  that “For GP data, mean±SD values were determined and utilized for two-tailed Student’s t-test analysis”. Indeed, we forgot to report the asterisks in the fig. 5C indicating a significant difference (p<0.05) of 16:0, 16:1_9 and 16:1_6 with respect to control. We apologize for this mistake.

A new Figure 5 is provided. Also Figure legend was modified accordingly:

Figure 5. Membrane fluidity of Caco-2 cells resulting from different fatty acid supplementations. (A)  ratiometric Laurdan fluidity maps of representative Caco-2 cells are reported after 4 hours for untreated cells (Ctrl), palmitic acid (16:0), palmitoleic acid (9cis-16:1) and sapienic acid (6cis-C16:1) 150 μM treatments in a two-colored pseudoscale, spanning from red (very fluid) to green (very rigid). Scale bar is 20 μm. (B) GP histograms of the fluidity images for (from top to bottom): untreated cells (Ctrl), palmitic acid (16:0), palmitoleic acid (9cis-C16:1) and sapienic acid (6cis-C16:1). The range of the histograms goes from 0 (very fluid regions) to 1 (very rigid regions). (C) Mean Values of GP histograms for the different treatments (*p<0.05 compared to Ctrl).

The authors suggested palmitoleic acid and sapienic acid treatment induced the biological difference. Whether these differences in lipid metabolism process lead to the biophysical differences in colon cancer cell lines? In what mechanism

The referee was right to raise this point. We amended the results (page 8, lines 276-279) to evidence the contents of unsaturated fatty acids in the case of sapienic acid, which make the difference. In the Discussion section (page 12, lines 438-444) we better explained the results in terms of different fluidity caused by the lipid metabolism. In page 12 Lines 453-454 we wrote: “The formation of n-10 MUFA, the PUFA sebaleic acid and the diminution of SFA can clearly contribute to the increase of fluidity observed in the cells clearly contribute to the increase of fluidity observed in the cells”.

FFA should be conjugate with BSA first when being added into the medium, did authors treat the cells in this way?  It should be described in the method part.

The addition of fatty acids can be done using stock solutions of fatty acids in ethanol as reported in several papers in the literature regarding the fatty acid supplementation in cell cultures, and also in our procedures published so far.

There are several careless mistakes in this manuscript. For example, Figure 4C should be 5C in result 2.5.

A careful check of the revised paper was done.

Round  2

Reviewer 2 Report

      The authors addressed all of the issues raised in the original review. There is no other concern.